# On the Physics of Kayaking

**Charlie Prétot** [1,2] **, Rémi Carmigniani** [1] **, Loup Hasbroucq** [2] **, Romain Labbé** [3] **, Jean-Philippe Boucher** [3] **and Christophe Clanet** [2,*]

1    LHSV, Ecole des Ponts, EDF R et D, 77455 Marne-la-Vallee, France
2    LadHyX, UMR 7646 du CNRS, Ecole Polytechnique, 91128 Palaiseau, France
3    Phyling, Drahi Xnovation Center, Ecole Polytechnique, 91128 Palaiseau, France
*    Correspondence: christophe.clanet@ladhyx.polytechnique.fr

**Abstract:** The propulsion force of a kayaker can be measured thanks to sensors placed on the paddle. This article aims at linking this force to the evolution of the velocity of the boat. A general model is proposed to describe the motion of a K1 kayak. To validate the model and evaluate the relevant physics parameters, three on-water kayaking trials are proposed: a pure deceleration, a standing start, and $10 \times 50$ m with two athletes at the national level. These trials were performed with a force sensor on the paddle and video recording. We used the deceleration to evaluate the drag of the boat. Then the standing start showed that there was an active drag coefficient while kayaking. Finally, the $10 \times 50$ m exhibited a power law of one-third between the velocity and the stroke rate. The acceleration during the standing start together with the relationship between the velocity and stroke rate were well captured theoretically. This approach enabled us to evaluate the important parameters to describe a kayak race: the drag of the boat, an active drag coefficient, the mean propulsive force, and a propulsive length. It can be used to characterize athletes and monitor their performances.

**Keywords:** propulsion; stroke rate; velocity; drag

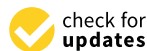



## 1. Introduction

In a kayak, the paddler is seated in the direction of motion (contrary to rowing) and uses a double-bladed paddle, as presented in Figure 1a–c. Kayak races are ruled by the International Canoe Federation [1], which states that two different disciplines exist at the summer Olympics: slalom in river and sprint on flatwater. The latter is the one studied in the present paper. K1, K2, and K4 indicate the number of kayakers per boat; we present in Figure 1d the nine different events that took place in Rio-2016 with the corresponding race lengths for both men and women and the different boat sizes and minimum weights. The time of the winner is indicated in red and the corresponding mean velocity is in blue.

Jackson [2], in his seminal work entitled *Performance prediction for Olympic kayaks*, gives the main characteristics of flatwater kayak races as far as physics is concerned: for single male and female races (K1), the boats of typical length $L = 5.2$ m reach $V = 4.83$ m/s for men and $V = 4.24$ m/s for women using paddles of blade areas $A_B = 0.063$ m$^2$ and a stroke cycle frequency of $f \approx 1$ Hz. Compared to the results of the Rio Olympic games presented in Figure 1b, we observe that these numbers are still of a good order of magnitude.

The underlying hydrodynamics has since been studied in detail for both hulls [3,4] and paddles [5,6], allowing the optimization of shapes. Using the length of the boat, $L$, and its velocity, $V$, we evaluated the Reynolds number $Re_L = \rho V \cdot L / \eta = 2.5 \times 10^7$ and the Froude number $Fr = V / \sqrt{gL} = 0.67$. Both values indicate that the total drag $F_D$ can be written as the sum of the turbulent skin friction and the wave drag [2]:

$$F_D = \frac{1}{2} \rho V^2 \left( S \cdot C_{sj} + \frac{\Omega^{5/3}}{L^3} C_{wj} \right), \tag{1}$$

where $S$ is the wetted area, $C_{sj} = 0.0028$, the turbulent skin drag coefficient, $\Omega$ is the immersed volume, $C_{wj} = 11$, the wave drag coefficient. Since the wetted surface is related to the immersed volume by the relation $S \approx 2.5\sqrt{\Omega L}$, one deduces for an athlete and boat mass of 90 kg at $V = 4.8$ m/s, a drag force $F_D \approx 75$ N, which means a dissipated power $P_D = F_D \cdot V \approx 360$ W. In this example, the wave drag (second term in Equation (1)) accounts for 22% of the total drag.

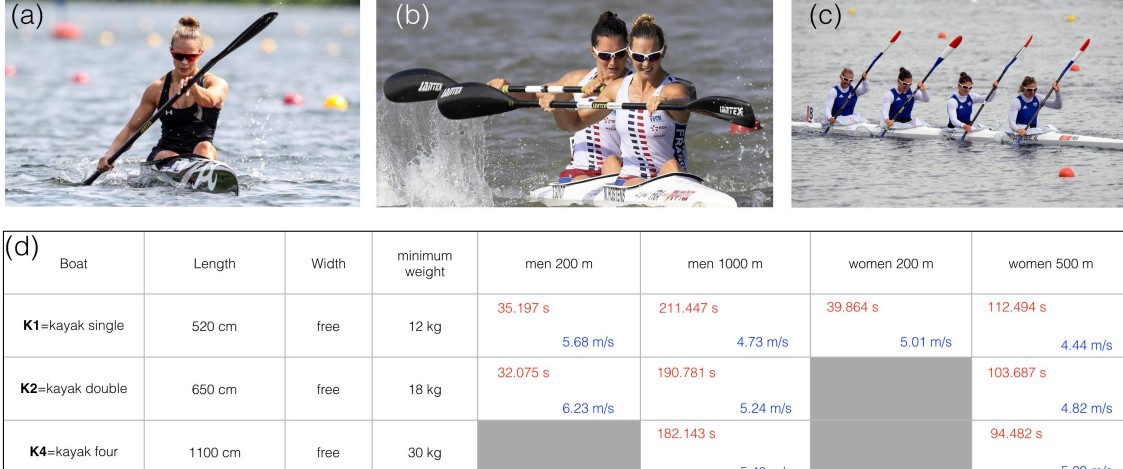

| (d) Boat | Length | Width | minimum weight | men 200 m | men 1000 m | women 200 m | women 500 m |
|---|---|---|---|---|---|---|---|
| **K1**=kayak single | 520 cm | free | 12 kg | 35.197 s <br> 5.68 m/s | 211.447 s <br> 4.73 m/s | 39.864 s <br> 5.01 m/s | 112.494 s <br> 4.44 m/s |
| **K2**=kayak double | 650 cm | free | 18 kg | 32.075 s <br> 6.23 m/s | 190.781 s <br> 5.24 m/s | | 103.687 s <br> 4.82 m/s |
| **K4**=kayak four | 1100 cm | free | 30 kg | | 182.143 s <br> 5.49 m/s | | 94.482 s <br> 5.29 m/s |

**Figure 1.** (**a**) K1 women's kayak race, (**b**) K2 women's kayak race, (**c**) K4 women's kayak race, (**d**) presentation of the 9 Olympic events in kayaks at the Rio 2016 Olympic games with corresponding boat characteristics. The time of the winner is indicated in red and the corresponding mean velocity in blue.

To our knowledge, in the existing literature on kayaking, few have directly measured the propulsion force in the paddle. Bjerkefors [7] studied the power output but on an ergometer whereas Klitgaard [8] showed that there are significant kinematic differences between the ergometer and the ecological situation. Most of the on-water trials focus on the kinematic measurements of the instantaneous velocity [9], or of the average stroke rate, stroke length, and velocity per 50 m [10]. The force on the paddle was evaluated for kayaking [11–14] and canoeing [15] but was not linked to the kinematics directly. On the contrary, Delgado [16] proposed a model for the evolution of the velocity of a kayak without experimental measurements of the force and the velocity.

In the present paper, we propose an on-water study with kinematics and dynamic measurements thanks to an instrumented paddle. Our goal here was to quantify and model the dynamics of the kayak through simple physics parameters. We could then compare the measured force and the evolution of the velocity thanks to the equation of motion. The paper is organized as follows. In the first section, we detail the experimental set-up and our theoretical motivations. Then we present three trials to validate and interpret our model: a pure deceleration trial, a standing start, and a $10 \times 50$ m with a progressive increase of the velocity between the trials. The first trial characterizes the drag of the boat. The last two trials validate the model in the transient regime and the steady regime.

## 2. General Model and Experimental Set-Up

### 2.1. Theoretical Motivation

In this article, we focus on the K1 kayak, meaning that there is only one athlete on the boat. The dynamics of the kayak are described by Newton's equation, which takes the following scalar form along the direction of motion:

$$M_e \frac{dV}{dt} = F_m(t) - F_D(t), \tag{2}$$

where $M_e = M_t + M_a$ is the sum of the total mass $M_t = M_k + M_b$ ($M_k$ is the mass of the kayaker and $M_b$ is the mass of the boat) and of the added mass $M_a$, which accounts for the mass of the water entrained by the boat. In Equation (2), the force $F_m(t)$ is the propulsive force exerted by the blade on the water in the direction of motion and $F_D(t)$ is the total drag.

We wanted to evaluate the different contributions of this equation and find a simple model of the evolution of the velocity in the kayak. In the first experiment, we set $F_m(t) = 0$. This pure deceleration trial allowed us to evaluate the drag of the kayak. In the second experiment, we performed a standing start, measuring $F_m(t)$ and linking this force to the evolution of the velocity. Finally, we set $M_e dV/dt = 0$ in steady motion at different velocities to exhibit a velocity–stroke rate relationship thanks to the balance of forces.

## 2.2. Experimental Set-Up

### 2.2.1. Athletes and Boats

The experiments were conducted at the nautic club of Ecole Polytechnique with two male expert kayakers who used their own kayaks. We named them A1 and A2. The main characteristics of the kayakers and their boats are presented in Figure 2. A1 is a national U23 athlete, of height 1.70 m and weight 64 kg. A2 is an international U23 athlete, height 1.81 m, and weight 74 kg. The athletes both used a Gamma Rio M blade from Jantex mounted on a Kevlight MFTech shaft equipped with strain gauges (see Figure 3). The athletes could vary the paddle lengths based on their preferences. The total mass of the paddle was 730 g and was neglected compared to the boat and athlete masses.

| athlete | | | boat | | paddle | | |
|---|---|---|---|---|---|---|---|
| name | height mass age | specialty level | type & brand | boat mass length maximal width | blade | Shaft | total length of the paddle |
| A1 | 1.70 m 64 kg 23 yo | Ocean Racing National U23 |  Quattro M - Nelo | 12 kg 5.20 m 0.41 m |  Gamma Rio M Jantex 0.076 m² | Kevlight MFTech | 2.16 m |
| A2 | 1.81 m 74 kg 23 yo | Sprint International U23 |  Wakatwo-KickTheWaves | 10 kg 4.50 m 0.61 m |  Gamma Rio M Jantex 0.076 m² | Kevlight MFTech | 2.10 m |

**Figure 2.** Main characteristics of the athletes, boats, and paddles used in the experiments.

### 2.2.2. Force Measurement

Strain gauges were placed on the paddle and linked to an acquisition card attached to the middle of the paddle (see Figure 3). The force was recorded at a sample rate of 100 Hz and the data were stored in a waterproof box *MaxiPhyling* placed in the boat. We used Bluetooth to connect the acquisition card and the *MaxiPhyling* box. In the same waterproof box, there was an accelerometer and a gyrometer, evaluating the contribution of the yaw, pitch, and roll. The data are stored on an SD card in the box. The force sensor was calibrated with calibration masses attached to the center of the immersed part of the paddle (Figure 3). Both sides were calibrated separately.

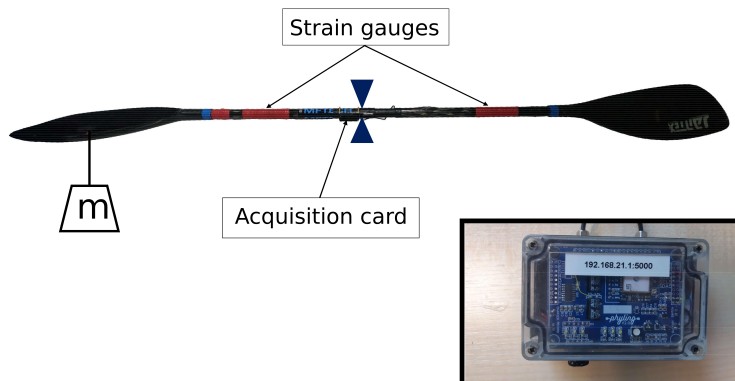

**Figure 3.** Calibration of the instrumented paddle used for the experiments. The mass "m" symbolizes the mass of calibration. On the right bottom corner, we show the waterproof box *MaxiPhyling* placed on the boat.

### 2.2.3. Velocity Measurement

We recorded the trials via a fixed camera (GoPro) set on a tripod placed on the lake bank. The sample rate was 60 frames per second. The selected angle of view made it possible to observe 40 m of the kayak displacement on the water. We tracked both the front extremity $(x_F, y_F)$ and back extremity $(x_B, y_B)$ of the boat. We call M the center of the boat $(x_M, y_M) = (\frac{x_B + x_F}{2}, \frac{y_B + y_F}{2})$. All the coordinates are defined Figure 4. Knowing the boat length $L_B$ (in meters), we could compute the velocity as:

$$V(t) = \frac{\sqrt{(x_M(t + \Delta t) - x_M(t - \Delta t))^2 + (y_M(t + \Delta t) - y_M(t - \Delta t))^2}}{\sqrt{(x_F(t) - x_B(t))^2 + (y_F(t) - y_B(t))^2}} \frac{L_{\text{boat}}}{2\Delta t} \quad (3)$$

The coordinates $(x, y)$ are in pixels. We used this formula to correct the effect of perspective along the trajectory. We checked that the length of the boat in pixels did not vary in the interval of time $[t - \Delta t, t + \Delta t]$. Depending on the experiment we performed, we used different values of $\Delta t$. This was a trade-off between filtering the noise of measurement and capturing the fast evolution of the velocity.

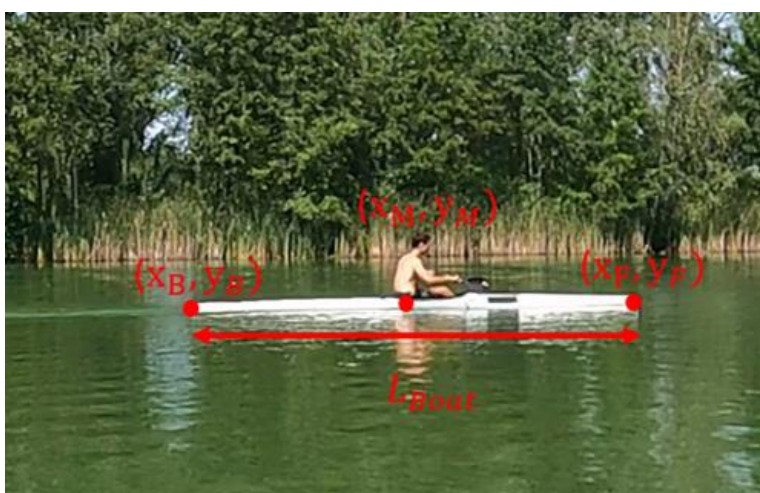

**Figure 4.** Description of the notation used to compute the velocity of the boat. The coordinates $(x, y)$ are in pixels whereas $L_{\text{boat}}$ is in meters.

### 3. Pure Deceleration: The Zero Propulsion Limit, $F_m(T) = 0$

Pure deceleration was achieved when the kayaker stopped paddling and removed the blades from the water, keeping a constant position along a straight path.

#### 3.1. Observations and Velocity Evolution

An example of a pure deceleration sequence is presented in Figure 5. After a few paddling cycles, which allowed the kayaker to reach the velocity $V_0$, the kayaker stopped paddling and kept the same position along a straight path down to rest.

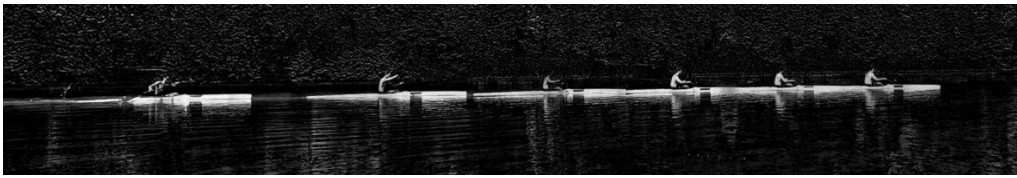

**Figure 5.** Chronophotography illustrating a deceleration test performed with L to measure the effective mass $M_e$ via the deceleration of the boat. The time between each frame was $\Delta t = 0.167$ s.

Quantitatively, the time evolution of the velocity during the deceleration is shown in Figure 6a. We observed that the decrease was not linear and that it took typically 5 s for the velocity to decrease by a factor of 2 from its initial value $V_0 \approx 4.5$ m/s to $V_0/2$. To describe this deceleration, we use the no-propulsion limit ($F_m = 0$) in Equation (2), which reduces to:

$$M_e \frac{\mathrm{d}V}{\mathrm{d}t} = -\frac{1}{2}\rho \, SC_D \, V^2, \tag{4}$$

where $SC_D$ is the total drag area. Assuming that $SC_D$ remains almost constant, the theoretical solution deduced from Equation (4) is:

$$\frac{V_0}{V(t)} = 1 + \frac{t}{\tau} \quad \text{where} \quad \tau = \frac{2M_e}{\rho \, SC_D \, V_0}. \tag{5}$$

In this expression, $\tau$ is the characteristic time over which the velocity decreases ($V(t = \tau) = V_0/2$). The corresponding time evolution of the velocity ratio $V_0/V(t)$ is presented in Figure 6b for Athlete A1 (black squares) and for Athlete A2 (red squares). In both cases, the theoretical affine relationship expected from Equation (5) was observed; we measure $1/\tau \approx 0.180 \pm 0.005$ s$^{-1}$. Thus, the velocity decreases over the characteristic time $\tau \approx 5.55$ s.

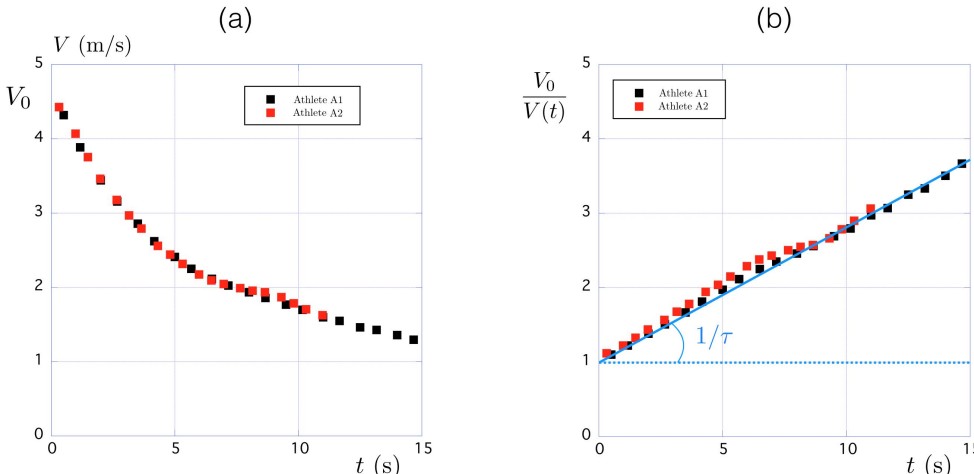

**Figure 6.** (**a**) Time evolution of the velocity $V(t)$ for Athlete A1 (black squares) and Athlete A2 (red squares). (**b**) Time evolution of the velocity ratio $V_0/V(t)$ for Athlete A1 ($V_0 = 4.76$ m/s) and Athlete A2 ($V_0 = 4.98$ m/s).

### 3.2. The Effective Mass $M_e$

As seen in Equation (2), the inertia of the kayaker and his boat did not only depend on $M_t$, the sum of the mass of the kayaker $M_k$, and the mass of the boat $M_b$. The added mass could be estimated using the data for an ellipsoid presented in Figure 7 [17]. The volume of the ellipsoid of the long axis $a$ and small axis $b$ is $\Omega_{el} = 4/3\pi ab^2$. While moving along the large axis direction, the added mass is a fraction of the displaced mass $M_a = K_{am}\rho\Omega_{el}$ where the constant $K_{am}$ depends on the aspect ratio $a/b$ as shown in Figure 7b. For a sphere ($a/b = 1$) one recovers the classical result $K_{am} = 1/2$. For a kayak of aspect ratio $a/b \approx 5.2/0.41 = 12.7$ (Athlete A1) one finds $K_{am} \approx 0.017$ so that the added mass can be estimated to $M_a = K_{am} \cdot M_t \approx 1.3$ kg. Thus, the corresponding effective mass for Athlete A1 is deduced: $M_e = 77.3$ kg. The same calculation for Athlete A2 leads to $K_{am} = 0.0332$ so that $M_a = K_{am} \cdot M_t \approx 2.8$ kg from which $M_e = 86.8$ kg.

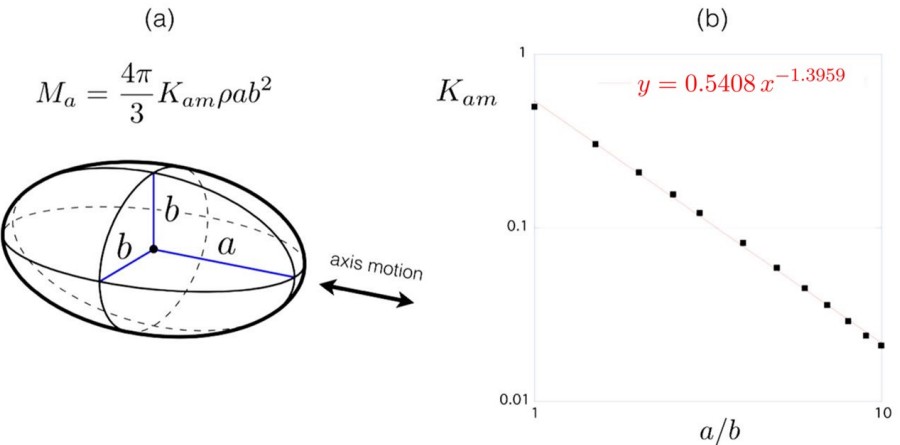

**Figure 7.** (**a**) Presentation of the ellipsoid. (**b**) Added mass factor $K_{am}$ as a function of the aspect ratio $a/b$.

### 3.3. The Total Drag, $F_d$

The total drag $F_D = 1/2\rho \, SC_D \, V^2$ is composed of three different contributions, the skin friction, $F_s$, on the slender hull associated with the immersed part of the boat; the wave drag, $F_w$, associated with the interfacial wake produced by the motion of the boat; and the aerodynamic drag, $F_a$, due to the airflow around the bluff kayaker [18,19]. Using the expression of the deceleration time $\tau$ in Equation (5), we can estimate the value of the total drag area $SC_D$ for the two kayakers. Using $\tau = 5.55$ s, $M_e = 77.3$ kg and $V_0 = 4.76$ m/s we find $SC_D = 5.85 \times 10^{-3}$ m$^2$ for Athlete A1. Using $\tau = 5.55$ s, $M_e = 86.8$ kg and $V_0 = 4.98$ m/s we find $SC_D = 6.38 \times 10^{-3}$ m$^2$ for Athlete A2. In Appendix A, we show that we can recover theoretically this value by estimating the contribution of each involved drag. In the present discussion, we neglected the impact of the wind and potential currents in the lake. As we show in Appendix A, aerodynamic drag accounts for 6% of the total drag. Therefore, we expect a small impact of the wind (typically 5 km/h) on the results. Concerning the currents, we performed the experiments in a small closed lake, and we considered that they were negligible.

## 4. Standing Start

In this section, we analyze the results of the standing start trial and recovered the evolution of the velocity thanks to the force measurement and the estimation of drag obtained in the previous section. Before analyzing the first couple of strokes, we analyzed a single stroke.

### 4.1. Single Paddling Stroke

An example of a single paddling stroke is presented in Figure 8. This stroke corresponds to the third stroke after the start in the present case (see blue arrow in Figure 9). The sequence in (a) decomposes the stroke from the entrance of the paddle (image 1) to its exit (image 12). The time lapse between images was constant ($\Delta t = 0.05$ s) so the whole stroke lasted 0.55 s. The angle $\theta$ between the paddle and the water surface is defined in Figure 8b together with the measured normal force $\underline{F}$.

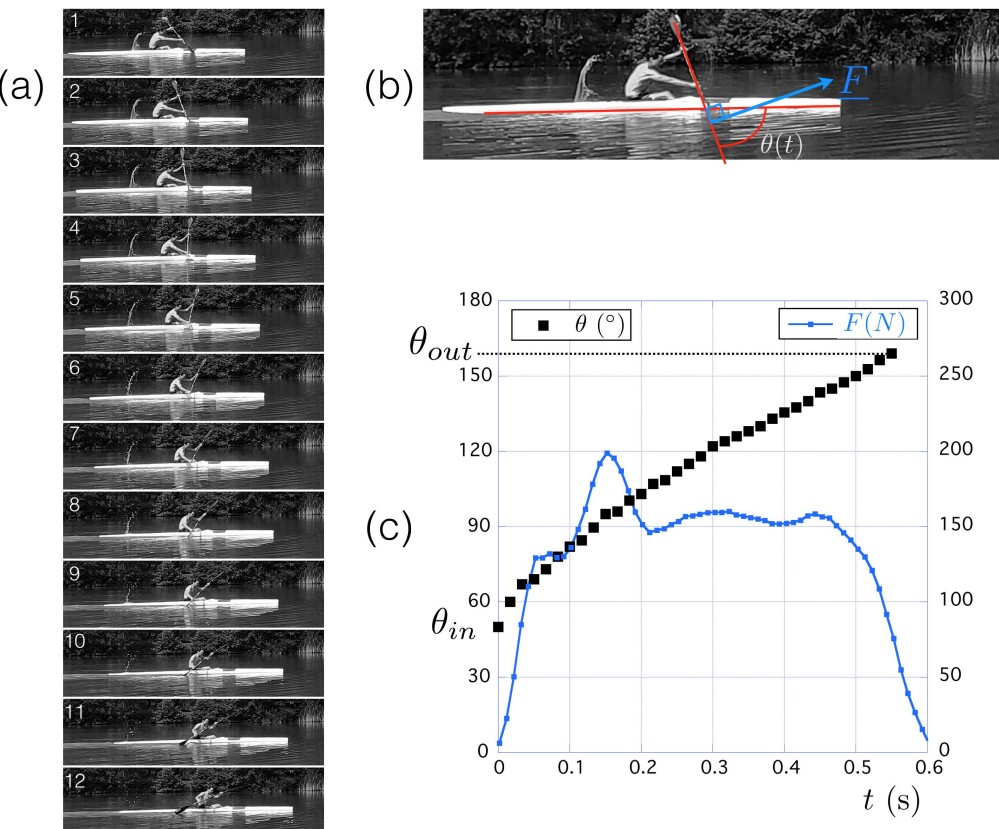

**Figure 8.** (**a**) Time sequence of a single paddling stroke performed by Athlete A1. The time lapse between each image is $\Delta t = 0.05$ s. (**b**) Definition of the paddle angle $\theta$ and the normal force $\underline{F}$. (**c**) Time evolution of the paddle angle $\theta$ (black squares) and the normal force intensity $F$ (blue squares).

The time evolution of $\theta$ and $F = \|\underline{F}\|$ during the stroke is shown in Figure 8c. Focusing on the paddle angle, we observe that this angle starts at $\theta_{in} \approx 50°$, quickly increases to 60°, and then evolves with an almost constant slope of $\omega_P = 3.2$ rad/s up to the exit angle $\theta_{out} \approx 160°$. Concerning the force, it increases during the entrance of the blade (first 0.1 s) and decreases during its exit (last 0.1 s) and exhibits a mean value of the order of 155 N in between. We observe little variations of $\theta_{in}$ and $\theta_{out}$ between the stroke cycles and the athletes between the different tests (less than 5° on the typical variation of 100°).

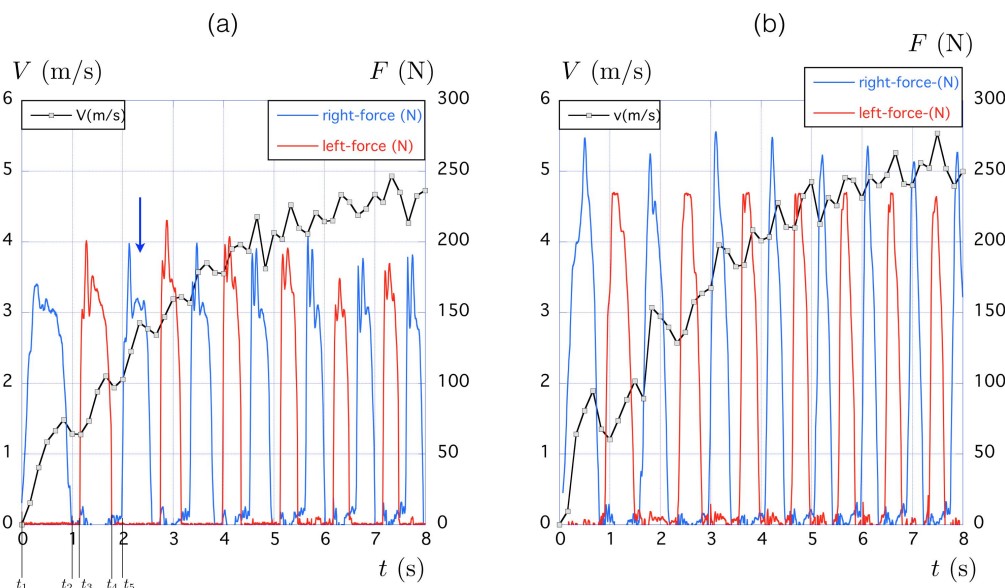

**Figure 9.** Time evolution of the velocity (solid black line with units on the left vertical axis) and the forces (solid colored curves with units on the right vertical axis) corresponding to the chronophotography presented in Figure 10. (**a**) Data for Athlete A1. The blue arrow indicates the cycle that is detailed in Figure 8. (**b**) Data for Athlete A2. The blue color is used for the force measured on the right paddle while the red color is used for the left. The quantity $F$ is the intensity of the force exerted in the direction normal to the paddle surface.

### 4.2. Experimental Data of the Standing Start

The two kayakers were asked to perform a standing start "as fast as they could" to characterize the acceleration phase. A chronophotography composed of the superposition of six pictures taken at equally spaced times $\Delta t = 1.66$ s is presented in [Figure 10a for Athlete A1 and Figure 10b for Athlete A2].

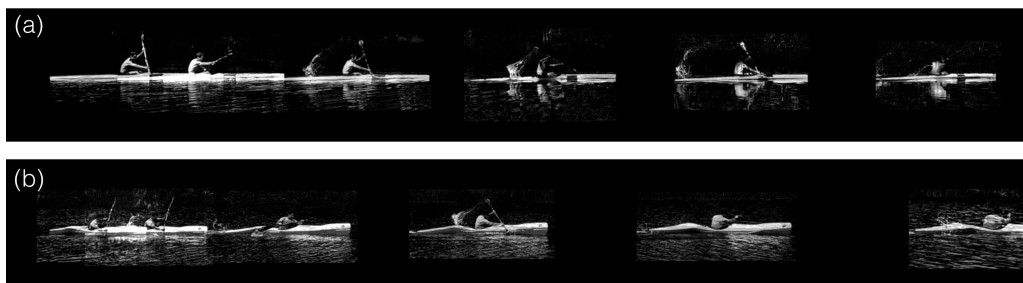

**Figure 10.** Chronophotography of a standing start. The time lapse between images is $\Delta t = 1.67$ s: (**a**) Athlete A1. The length scale is given by the length of the boat (5.2 m). (**b**) Athlete A2. The length scale is given by the length of the boat.

The corresponding time evolution of the velocity is presented with black solid lines in Figure 9 for the two kayakers. The grey squares underline that the velocity was measured every 0.16 s at a frequency of 6 Hz. Qualitatively, the velocity build-ups were similar for the two athletes and we first analyzed the one obtained with Athlete A1 (Figure 9a). The velocity increased from 0 to 5 m/s over a characteristic time of the order of 5 s. The associated acceleration is thus of the order of 1 m/s². The fluctuations observed on the velocity signal are associated with the periodic motion of the paddle. This is shown with the force signal reported in blue for the right blade and in red for the left blade. The maximal values of the blade force are of the order of 200 N.

Even if the maximal forces were larger for the second kayaker (Figure 9b), the same features were observed in the time evolution of his velocity. This observation reveals that

even if the boat and kayaker specialties are different, as underlined in Figure 2, the dynamics of the boat during a standing start are characterized by generic features, which need to be studied.

We define the period $T$ as the time needed for a blade to perform a full cycle, from one water entry to the next. The stroke rate $f$ is defined as $1/T$. In Figure 9, we show for the first stroke the different time markers. A period corresponds to $T = t_5 - t_1$. We observe in Figure 9 that the first cycle was longer than the subsequent ones for both kayakers. We also observe that there was a delay with no force between the end of a propulsive phase and the beginning of the next one. This corresponds to the recovery time $t_0$ during which both paddles were in the air. During all of those phases, the velocity of the boat systematically decreased due to the friction with the water and air. For the first cycle, $t_0 = (t_3 - t_2) + (t_5 - t_4)$.

Quantitatively, the evolution of the (period $T$) and of the recovery time $t_0$ are presented in Figure 11 as a function of the cycle number $n$: for the first kayaker, the period decreased quickly from 2 s in the first cycle to 1 s in the fifth cycle. The decrease was quick in the sense that, in the second cycle, the period was already 1.34 s (Figure 11a). Concerning the recovery time $t_0$, it remained almost constant and equal to $t_0 \approx 0.31$ s. In the last cycles, 2/3 of the period was dedicated to the propulsive phase and 1/3 to the side-change (aerial phase).

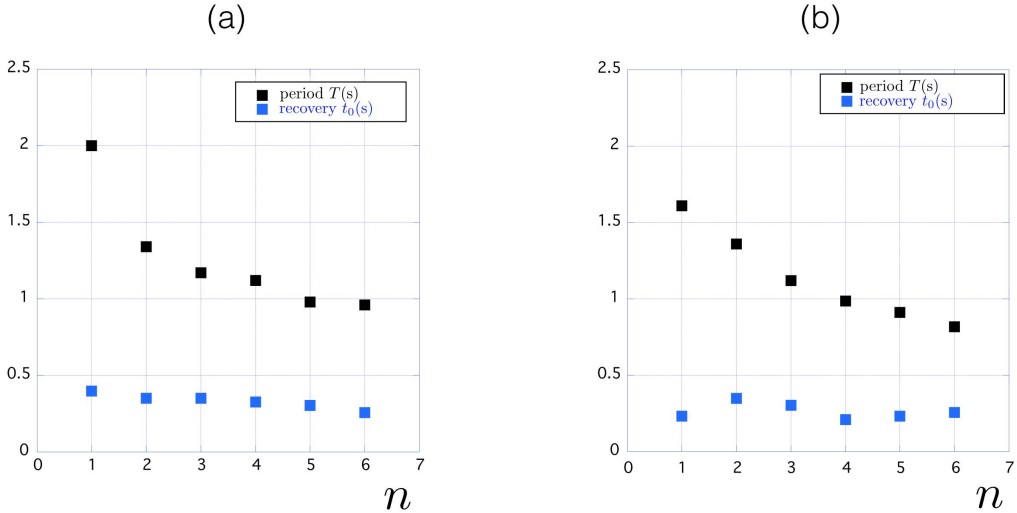

**Figure 11.** Evolution of the period $T$ and of the recovery time $t_0$ as functions of the cycle number $n$ during the two standing starts presented in Figure 9: (**a**) Athlete A1, (**b**) Athlete A2.

For the second kayaker, the evolution is presented in Figure 11b. The period also decreased by a factor of 2 between the initial cycle ($T = 1.6$ s) and the sixth cycle ($T = 0.8$ s) but the evolution was slower. The recovery time $t_0$ remained almost constant and equal to $t_0 \approx 0.28$ s. Again, in the last cycles, we recover the 2/3–1/3 proportions between the propulsive and the aerial phases.

*4.3. Theoretical vs. Experimental Velocity*

Knowing the effective mass $M_e$ (Section 3.2) and the total drag $F_D$ (Section 3.3) one can use the full Equation (2) to predict the time evolution of the velocity for a given propulsion force $F_m(t)$. This was done by solving the differential equation:

$$M_e \frac{\mathrm{d}V}{\mathrm{d}t} = F(t)n\theta(t) - K\frac{1}{2}\rho S C_D V^2 \quad \text{with} \quad \theta(t) = \theta_{in} + (\theta_{out} - \theta_{in})\frac{t - t_{in}}{t_{out} - t_{in}} \tag{6}$$

For the standing starts presented in Figure 9, this equation was solved with the initial condition $V(t = 0) = 0$ together with the measured force $F(t)$. The paddling dynamics are accounted for via the entrance ($\theta_{in}$) and exit angles ($\theta_{out}$) together with the corresponding

instants ($t_{in}$ and $t_{out}$). In the above Equation (6), we also introduced a constant $K \geq 1$ in front of the total drag $F_D = 1/2\rho S C_D V^2$ in order to account for the increase in the drag associated with the perturbed motion of the boat (surge, heave, sway, pitch, yaw, roll) that appeared once paddling started [1,20]: an active drag coefficient.

We compare in Figure 12 the velocity $V(t)$ obtained by the numerical integration of Equation (6) to the value measured experimentally. The comparison for Athlete A1 is presented in Figure 12a and the one for Athlete A2 in Figure 12b. In both cases, the zero perturbation limit ($K = 1$) is shown with a blue solid line and the best fit with a red solid line. We obtained $K = 1.25$ for Athlete A1 and $K = 1.1$ for Athlete A2, which means that the drag while paddling was 25% larger than the one measured with no paddling for Athlete A1 and 10% larger for Athlete A2. This difference is probably associated with the stability of the boat, which was larger for Athlete A2. Indeed, the boat for Athlete A2 was wider ($w = 61$ cm) than the boat of Athlete A1 ($w = 41$ cm). This hypothesis is confirmed by the gyroscope placed in the boat. The average norm of the angular velocity $\overline{\omega} = \frac{1}{T} \int_0^T \sqrt{\omega_x^2 + \omega_y^2 + \omega_z^2} dt$ reached $43°$/s for Athlete A2 and $\overline{\omega} = 33°$/s for athlete A1 at high velocities. This shows the lack of stability of the most narrow boat of A1; however, this higher angular velocity can also be caused by a less efficient technique of A1. There should be an optimal width of the boat for a given kayaker, minimizing the coefficient $KSC_D$.

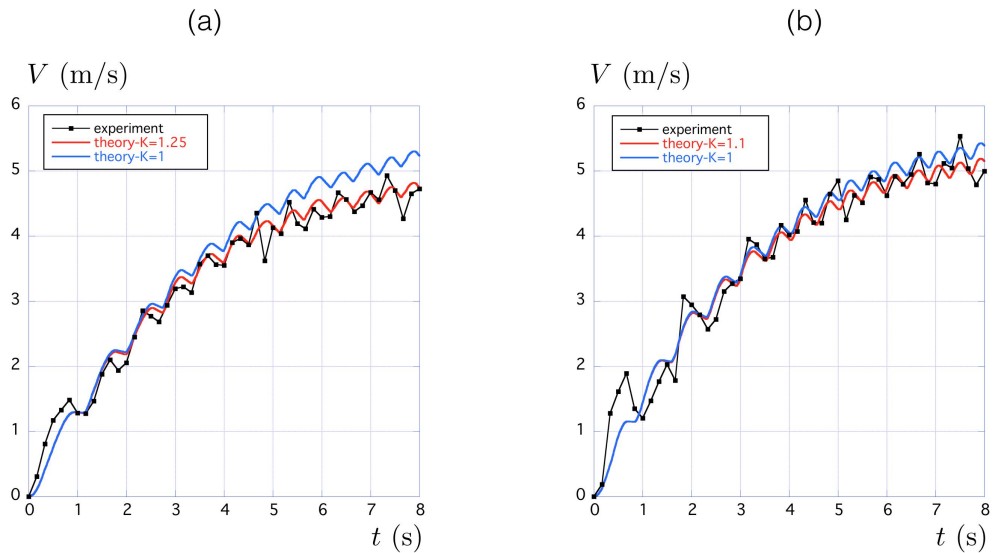

**Figure 12.** Comparison between the velocity measured experimentally and the one obtained by the numerical integration of Equation (6), for Athlete A1 (**a**) and for Athlete A2 (**b**).

### 4.4. An Algebraic Approximate Solution for the Mean Velocity

A different way to look at the motion of the kayak consisted of taking a picture of each time the right paddle entered the water. When such a move was done, we observed a steady motion where all the quantities were averaged over a period: $\overline{\zeta} = 1/T(n) \int_{t(n)}^{t(n)+T(n)} \zeta(t) dt$ where $T(n)$ is the duration of the $n$th cycle and $t(n)$ is the time at which the $n$th cycle starts. The average total force $\overline{F}$ and average propulsive component $\overline{F_m} = \overline{Fn\theta}$ for the standing start (Figure 9) are shown in Figure 13a. We observe that these forces remain almost constant over the first seven cycles.

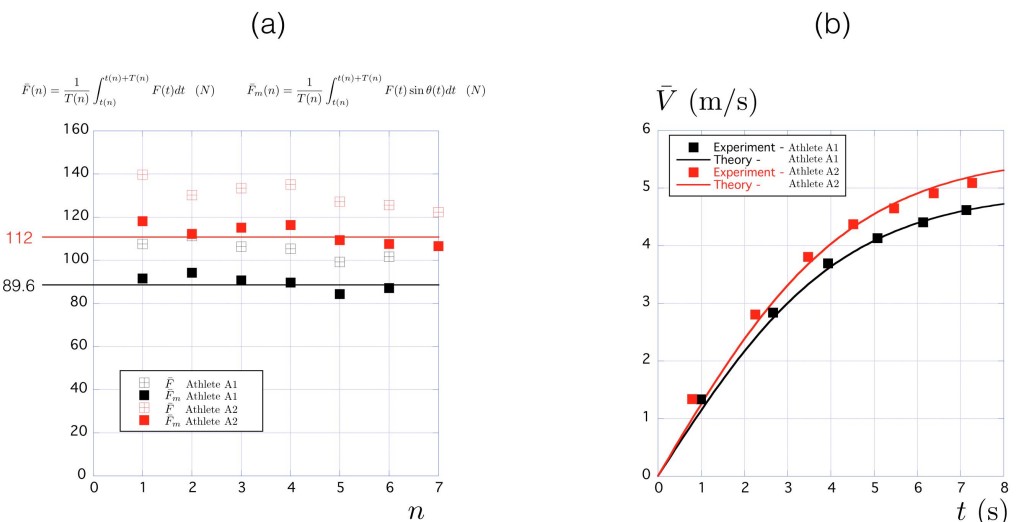

**Figure 13.** (**a**) Evolution of the mean total force $\overline{F}$ and mean propulsive component $\overline{F}_m = \overline{Fn\theta}$ over the first $n$ cycles of the standing start are presented in Figure 9 (**b**).

The equation of motion for the average quantities can be obtained by averaging Equation (6):

$$M_e \frac{\mathrm{d}\overline{V}}{\mathrm{d}t} = \overline{F}_m - K\frac{1}{2}\rho SC_D \overline{V}^2 \tag{7}$$

In this equation, we used the approximation $\overline{V^2} \approx \overline{V}^2$, which is valid if period $T$ was small compared to the characteristic time in which the velocity changes $1/\tau_c = 1/V\mathrm{d}V/\mathrm{d}t$. In the present application, the period is typically 1 s while the velocity changes over $\tau_c \approx 5$ s. Both values justify the approximation. Since $\overline{F}_m$ is shown in Figure 13a to also be constant, Equation (7) can be integrated and lead to the algebraic solution:

$$\overline{V}(t) = \overline{V}_{\max}\tanh(t/\tau) \quad \text{with} \quad \overline{V}_{\max} = \sqrt{\frac{2\overline{F}_m}{K\rho SC_D}} \quad \text{and} \quad \tau = \frac{M_e \overline{V}_{\max}}{\overline{F}_m}, \tag{8}$$

where $\tau$ is the characteristic time needed to reach the maximal velocity $\overline{V}_{\max}$. Using the values obtained for Athlete A1 ($\overline{F}_m = 89.6$ N, $M_e = 77.3$ kg, $K = 1.25$, $SC_D = 5.85 \times 10^{-3}$ m$^2$) we have $\tau = 4.2$ s and $\overline{V}_{\max} = 4.95$ m/s. The same evaluation for Athlete A2 ($\overline{F}_m = 112$ N, $M_e = 86.8$ kg, $K = 1.1$, $SC_D = 6.38 \times 10^{-3}$ m$^2$) leads to $\tau = 4.4$ s and $\overline{V}_{\max} = 5.6$ m/s.

The comparison between the time evolution of the mean velocity $\overline{V}(t)$ measured experimentally and the algebraic solution (8) is presented in Figure 13b with a fair agreement for both kayakers.

## 5. The 10 $\times$ 50 Meter Trial: Kayaking at Constant Velocity

### 5.1. Experimental Data

In the third type of experiment, the athletes were asked to keep a constant velocity of over 50 m during 10 different trials. The constraint was to increase the velocity at each trial from the lower (at trial number 1) to the fastest velocity (in trial number 10). Stroke rate $f$ was not imposed and was measured afterward from the recorded movies. The relationship between the mean velocity $\overline{V}$ over 50 m and stroke rate $f$ is presented in Figure 14. For both athletes, we observed the same relationship: $\overline{V} = Af^{1/3}$ with $A \approx 4.4$.

In Figure 14b, we present the evolution of the mean projected force over the propulsion time $\widetilde{F} = \int_{t_{in}}^{t_{out}} F(t)n\theta(t)\mathrm{d}t/(t_{out} - t_{in})$. $\widetilde{F}_L$ and $\widetilde{F}_R$ correspond, respectively, to the force generated by the left arm and by the right arm. The evolution $\theta(t)$ was obtained thanks to the movie of the trial, whereas $F(t)$ was given by the sensor on the paddle. We plotted the mean value of $\widetilde{F}_L$ and $\widetilde{F}_R$ in the zone between 25 and 45 m. The values of the propulsive

force for the right arm and the left arm were close. After an increase in the three first trials, $\widetilde{F}$ remained almost constant during the seven last trials. We can express the mean force over a cycle $\overline{F} = f\left(\widetilde{F}_L(t_{out} - t_{in})_L + \widetilde{F}_R(t_{out} - t_{in})_R\right)$. By assuming that the kayaker propelled symmetrically, which is in fair agreement with the data, we can simplify the previous expression: $\overline{F} = 2\widetilde{F}(t_{out} - t_{in})/T$. We only present the results for kayaker A1, but the results are similar for A2.

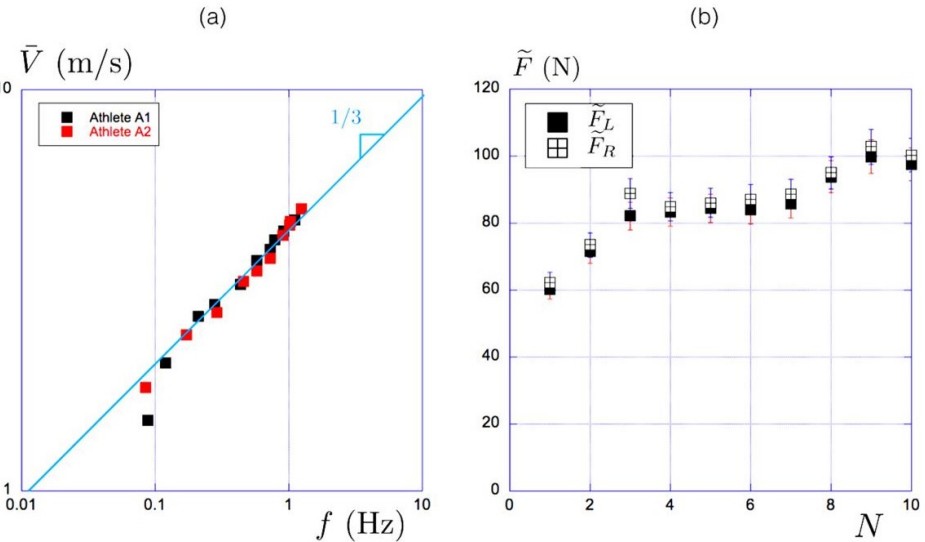

**Figure 14.** (**a**) Velocity–stroke rate relationship in the steady state regime for the two kayakers. (**b**) Evolution of the average propulsion force in the direction of paddling with trial N for Athlete A1. The force is averaged over the propulsion time.

### 5.2. Velocity–Stroke Rate Relationship Model

In this section, we model the relationship observed during the progressive $10 \times 50$ m: $\overline{V} = Af^{1/3}$ (Figure 14a). We start from Equation (7), corresponding to the average of Equation (2) over one cycle. Once the permanent regime is reached, $\frac{d\overline{V}}{dt} = 0$ (as $V(t) = V(t + T)$):

$$\overline{F_m} = K\frac{1}{2}\rho SC_D \overline{V}^2. \tag{9}$$

Using $t_p = t_{out} - t_{in}$, the propulsion time, as there are two propulsion phases (left and right assumed symmetric) during one cycle, Equation (9) yields:

$$\frac{2t_P}{T} \cdot \frac{1}{t_P} \int_{t_{in}}^{t_{out}} F(t)n(\theta(t))dt = K\frac{1}{2}\rho SC_D \overline{V}^2. \tag{10}$$

As observed in Figure 14, the mean projected force along the axis of the motion $\widetilde{F} = \int_{t_{in}}^{t_{out}} F(t)n(\theta(t))dt/t_P$ is approximately constant, except for the first two trials. Therefore, we define $\widetilde{F}_0$ as this constant value. As the stroke rate $f$ is the inverse of the period, we obtain:

$$f \cdot t_P \widetilde{F}_0 = K\frac{1}{4}\rho SC_D \overline{V}^2. \tag{11}$$

At this stage, we need to find an expression for the evolution of $t_P$ as a function of the velocity $\overline{V}$. The ratio between the hydrodynamic coefficients $SC_D$ of the boat and the paddle is small ($SC_D = 6 \times 10^{-3}$ m$^2$ for the boat and $SC_D = 1.2 \times 10^{-1}$ m$^2$ for a paddle). For this reason, we expect little drift of the paddles in the water compared to the boat during the propulsion. In the limit where the paddle is anchored in the water, the velocity of the paddle in the frame of the boat $V_{P/B}$ equals the velocity of the boat $\overline{V}$. Defining $L_p$ as the length of the propulsive path of the paddle in the frame of the boat, it comes $V_{P/B} = L_p/t_P$. Therefore, the relationship between $\overline{V}$ and $t_P$ should be close to $1/t_P = \overline{V}/L_p$. We used the

data of the $10 \times 50$ m of Athlete A1 to analyze the evolution of the inverse of the propulsion time and the mean velocity (Figure 15).

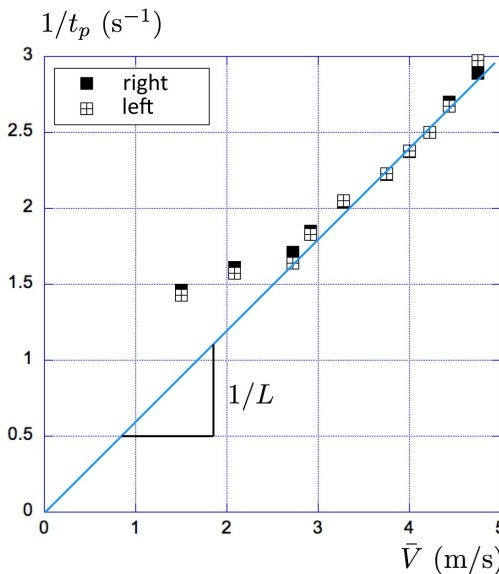

**Figure 15.** Evolution of $1/t_P$ as a function of $\overline{V}$ for Athlete A1 while performing $10 \times 50$ m.

In Figure 15, the relationship between $1/t_P$ and $\overline{V}$ is linear for velocities larger than 2 m/s for both arms. We found a propulsive length of $L_p = 1.64$ m for Athlete A1. This length can be compared to the arc length described by the center of a paddle, where we expect the force resultant to be applied. For Athlete A1, the distance between the two paddle centers was $\ell \approx 1.76$ m and we measured paddle angles going from $50°$ to $160°$. This corresponds to a traveled distance $L_p \approx 0.5\ell \times \Delta\theta \approx 1.69$ m. This represents a 3% difference with the previous measurement. Using the relation between $t_P$ and $\overline{V}$, Equation (11) can be simplified:

$$\overline{V} = \left( \frac{4L_p \widetilde{F}_0}{K\rho SC_D} \right)^{\frac{1}{3}} f^{\frac{1}{3}}. \tag{12}$$

This equation is compatible with the experimental data presented in Section 5.1. We found A = 4.2 for Athlete A1 ($\widetilde{F}_0 = 86$ N, $L_p = 1.64$ m, $K = 1.25$, $SC_D = 5.85 \times 10^{-3}$ m$^2$). This value must be compared to A = 4.4 found experimentally. Thus, we recovered the experimental behavior with a precision of 5%. In this section, we assumed symmetry between the left and right propulsion. In the standing start experiments (Figure 9), we observed a difference in the patterns between the left and right propulsion. However, if we focus on Figures 14b and 15, we observe that the propulsion time is the same for the left arm and the right arm and there is no significant difference in the mean value of the propulsion force between the left and right. As this model is averaged on a cycle, this is no issue that the force distribution over a cycle is not symmetric, while there are similar average values of the propulsion force and propulsion time.

However, we can also adapt the model in the case of a non-symmetric propulsion. In this case, in Equation (12) we replace $2L_p\widetilde{F}_0$ with $L_r\widetilde{F}_r + L_l\widetilde{F}_l$, where $L_r$ and $L_l$ are the right and left propulsive lengths and $\widetilde{F}_r$ and $\widetilde{F}_l$ are the right and left mean projected forces. This model has two additional parameters, which are not compatible with our wish to have a model as simple as possible. For this reason, we showed the model with the assumption of symmetry.

## 6. Conclusions

In the present paper, we provided methods to quantify and model the dynamics of kayak races. Three tests were used to evaluate the important physics parameters: a pure

deceleration, a standing start, and a progressive test of $10 \times 50$ m. The drag coefficient was evaluated from the first one. A typical value of $SC_D = 6 \times 10^{-3}$ m$^2$ was found. Theoretically, we were able to quantify the contribution of each term (skin, wave, and aerodynamic drag) and recover this value. The second test enabled us to quantify the effect of motion on the drag: an active drag coefficient. We found a drag increase of 25% ($K = 1.25$) and 10% ($K = 1.10$) for our two kayakers. The last test enabled us to evaluate the link between the velocity and stroke rate. For both kayakers, we found that this relationship could be written as $\overline{V} = A f^{1/3}$ with an explicit expression of parameter A for each athlete matching the experimental value.

The present work provides a general model to describe the propulsion in a kayak. This model is valid for the transient regime and the steady regime and could be used to develop a general optimization race algorithm coupled with physiological models [21].

The general approach proposed in the present work could be applied to other paddle-based sports, such as canoeing, rowing, and even swimming. It also presents a way to monitor and characterize athletes through four physics parameters: a drag parameter ($SC_D$), an active drag coefficient ($K \geq 1$), a mean projected force ($\widetilde{F}_0$), and a propulsive length ($L_p$).

**Author Contributions:** All the authors contributed equally to the paper. R.C. developed the model for the progressive test and wrote part of the paper, L.H. was in charge of analysing the data and collected part of the data, C.P. conducted the experiments, collected the data and wrote part of the paper, R.L. and J.-P.B. constructed the instrumented paddle and collected part of the data, C.C. developed the model for the standing start and wrote part of the paper. All authors have read and agreed to the published version of the manuscript.

**Funding:** This research was funded by Agence Nationale de la Recherche (ANR) grant number ANR-2020-STHP2-0006.

**Institutional Review Board Statement:** Both athletes volunteered for this study and gave their written consent to participate. Measurements were done as part of their usual training with no added task.

**Informed Consent Statement:** Informed consent was obtained from all subjects involved in the study.

**Data Availability Statement:** This article has no additional data.

**Acknowledgments:** We deeply thank the two athletes for taking part in the field experiments. We also thank Commandant Marc Mander for his interest in the Sciences 2024 program and for letting the kayakers use the Ecole Polytechnique rowing facility center. We finally thank Alexandre Rosinski for his technical help during the different test sequences.

**Conflicts of Interest:** The authors declare no conflict of interest.

## Appendix A

The objective of this section is to theoretically recover these experimental values and to estimate the relative contributions of the three forces $F_s$, $F_w$, and $F_a$.

### Appendix A.1. Skin Friction $F_s$

Skin friction $\underline{F}_s$ is opposed to the motion and its magnitude can be written as $F_s = 1/2\rho S C_s V^2$ where $S$ is the wetted surface and $C_s$ is the skin friction coefficient. This coefficient depends on the Reynolds number $Re_L = \rho V L/\eta$ based on the length of the hull $L$ (Figure A1) and the fluid density $\rho$ and dynamic viscosity $\eta$ (for water $\rho = 10^3$ kg/m$^3$ and $\eta = 10^{-3}$ Pa.s). In the range $10^4 < Re_L < 10^6$, the laminar coefficient is evaluated at $C_s = 1.33/\sqrt{Re_L}$ while at a larger Reynolds number, the commonly accepted formula is the one from the International Towing Tank Conference (ITTC) [18,22]: $C_s = (1+k)\, 0.075/(\log Re_L - 2)^2$. In this expression, $(1+k) = 2.76\left(L/\Omega^{1/3}\right)^{-0.4}$ is the shape factor, which depends on the length of the boat $L$ and the immersed volume $\Omega$ [23].

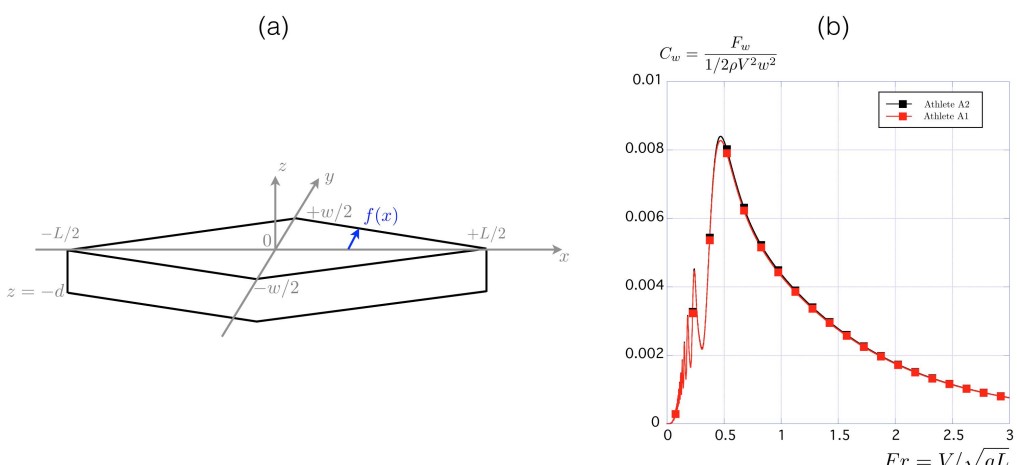

**Figure A1.** (**a**) Scheme of a prismatic hull. (**b**) Evolution of the wave drag coefficient $C_w$ with the Froude number calculated with Equation (A4).

In kayaking, the typical velocity is $V = 5$ m/sand the hull length $L = 5$ m, so that $Re_L = 2.5 \times 10^7$. Using $\Omega = 80 \times 10^{-3}$ m$^3$, one finds $(1 + k) = 1.035$. With this Reynolds number and shape factor, the above formula leads to the value $C_s = 0.0027$, which is not far from the value of 0.0028 estimated by Jackson [2]. Since the wetted surface is related to the immersed volume by the relation $S = 2.5\sqrt{\Omega L}$ [2], we finally get

$$F_s = \frac{1}{2}\rho SC_s V^2 \quad \text{with} \quad SC_s = \frac{0.5175L^2}{(\log Re_L - 2)^2}\left(\frac{\Omega}{L^3}\right)^{0.633}. \tag{A1}$$

For a given load $\Omega = M_t/\rho$, since the log term remains almost constant, we deduce that the skin friction increases with the square root of the length $L$ and almost quadratically with the velocity. For a typical load $M_t = 76$ kg, a boat length $L = 5.2$ m, and a boat speed $V = 5$ m/s the above Equation (A1) leads to $SC_s = 4.08 \times 10^{-3}$ m$^2$ or $F_s = 51$ N for Athlete A1. The same calculation for Athlete A2 ($M_t = 84$ kg, $L = 4.5$ m, $V = 5$ m/s) leads to $SC_s = 4.38 \times 10^{-3}$ m$^2$ or $F_s = 54.8$ N.

*Appendix A.2. The Wave Drag $F_w$*

The wave drag has a long history but one of the more compact forms is given by Mitchell formula [24–26]:

$$F_w = \frac{4}{\pi}\frac{\rho g^2}{V^2}\int_1^\infty \left(I^2 + J^2\right)\frac{\lambda^2 d\lambda}{\sqrt{\lambda^2 - 1}}, \tag{A2}$$

where

$$I = \int\int \frac{df}{dx}e^{\lambda^2 gz/V^2}\cos\left(\lambda gx/V^2\right)dxdz \tag{A3}$$

with a similar integral for $J$ involving sine instead of cosine. In this expression of the wave drag, the function $f(x, z)$ stands for the hull shape as presented in Figure A1a.

With a simplified linear profile invariant in $z$ (Figure A1a), the function $f$ takes the form $f(x) = w/L(x + L/2)$ for $x \in [-L/2, 0]$ and $f(x) = -w/L(x - L/2)$ for $x \in [0, +L/2]$. In this limit, the above formula for the wave drag reduces to :

$$F_w = \frac{1}{2}\rho V^2 w^2 \frac{128}{\pi}Fr^4 \int_1^\infty \left(1 - e^{-\lambda^2/(\beta Fr^2)}\right)^2 n^4\left(\frac{\lambda}{4Fr^2}\right)\frac{d\lambda}{\lambda^4\sqrt{\lambda^2 - 1}} \tag{A4}$$

where $\beta = L/d$ is the ration between the length of the boat $L$ and the draft $d$. For the simplified prismatic hull presented in Figure A1a, $\Omega = 1/2Lwd$ so that $\beta = 1/2L^2w/\Omega$. Using the values of the parameters $L$, $w$, and $M_t$ presented in Figure 2, we calculate the

wave drag coefficient $C_w = F_w / (1/2\rho V^2 w^2)$ and present its evolution as a function of the Froude number in Figure A1b.

The non-monotonic evolution is classical for the wave drag as well as its maximum value achieved for $Fr \approx 0.5$ [19,26]. Quantitatively for a 5.2 m long boat moving at $V = 5$ m/s one finds $Fr = 0.70$ and deduces $C_w = 6.1 \times 10^{-3}$, which leads to $F_w = 12.8$ N with $w = 0.41$ m (these values correspond to the one of Athlete A1 in Figure 2).

This value for the wave drag can be compared to the one proposed by Jackson using Equation (1): $F_{wj} = 1/2\rho V^2 C_{wj} \Omega^{5/3}/L^3$. For the same velocity, $V = 5$ m/s one finds $F_{wj} = 13.3$ N, which is in fair agreement.

The corresponding wave drag area $w^2 C_w$ is $1.02 \times 10^{-3}$ m$^2$ for Athlete A1 (or $F_w = 12.75$ N). The same calculation for Athlete A2 leads to $w^2 C_w = 2.09 \times 10^{-3}$ m$^2$ (or $F_w = 26.1$ N), the main difference being the width of the boat.

*Appendix A.3. The Aerodynamic Force $F_a$*

The last component of the drag is the aerodynamic contribution $F_a$, which is the air friction on the emerged bluff body composed of the upper part of the kayaker and of the paddle (Figure A2a). Using $gma_1$ for the frontal area of the kayaker and $gma_2$ for the unshadowed paddle, we have (without wind):

$$F_a = \frac{1}{2}\rho_a(gma_1 C_{D1} + gma_2 C_{D2})V^2, \tag{A5}$$

where $\rho_a$ is the air density and $C_{D1}$ (respectively, $C_{D2}$) is the drag coefficient associated with the frontal area $gma_1$ (respectively, $gma_2$).

(a)                                                           (b)

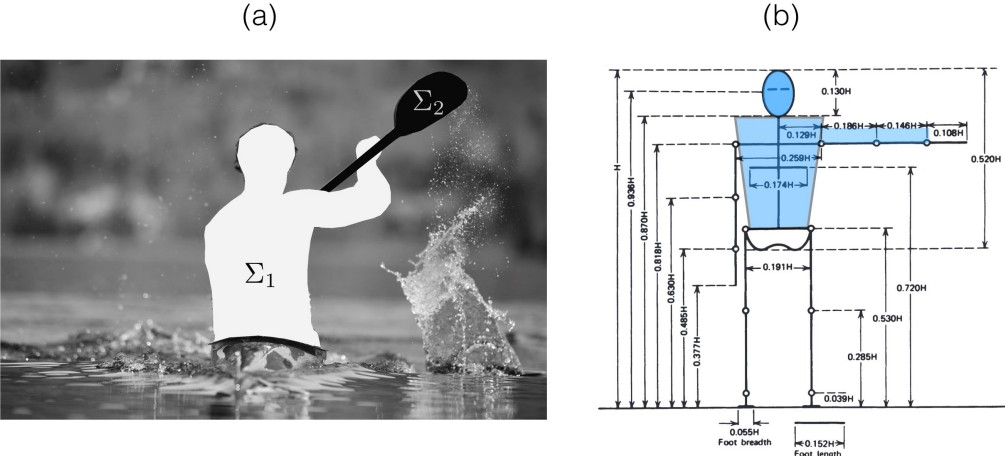

**Figure A2.** (**a**) Identification of the two bluff-emerged surfaces $gma_1$ for the kayaker and $gma_2$ for the unshadowed part of the paddle. (**b**) Anthropometric figure with data extracted from the work by Drillis and Contini; [27].

The frontal area of the body $gma_1$ can be estimated using the anthropometric data from Drillis and Contini (Figure A2b) as the blue area, which leads to $gma_1 = 0.101H^2$ where $H$ is the size of the kayaker. The associated drag coefficient is $C_{D1} = 0.7$ [28]. For the paddle, we measure $gma_2 = 0.08$ m$^2$. Using $C_{D2} = C_{D1}$ we thus estimate for $V = 5$ m/s, $F_a \approx 3.9$ N for Athlete A1, and $F_a \approx 4.3$ N for Athlete A2. The tables that we used correspond to the average distribution of mass for the population. The kayakers have more developed muscles in the upper body than in the lower body, which is why we should have slightly underestimated the aerodynamic force.

*Appendix A.4. Comparison between the Theoretical Drag and the One Measured by Deceleration*

The theoretical total force at $V = 5$ m/s is thus $F_D = F_s + F_w + F_a = 51 + 12.75 + 3.9 = 67.7$ N for Athlete A1, which corresponds to $SC_D = 5.4 \times 10^{-3}$ m$^2$. This value is 7.5%

smaller than the one measured via the deceleration ($SC_D = 5.85 \times 10^{-3}$ m$^2$). Considering the relative importance of the different contributions of the drag, we find here 75% for skin friction, 19% for the wave drag, and 6% for the air drag. Even if we slightly underestimated the aerodynamic force in the last section, this is not the major part of the total drag.

For Athlete A2, the theoretical total force at $V = 5$ m/s is $F_D = F_s + F_w + F_a = 54.8 + 26.1 + 4.3 = 85.2$ N. This corresponds to $SC_D = 6.8 \times 10^{-3}$ m$^2$. This value is 6.7% larger than the one measured via the deceleration ($SC_D = 6.38 \times 10^{-3}$ m$^2$). The proportions of the three contributions are 64% for skin friction, 30% for the wave drag, and 6% for the air.

The main difference with Athlete A1 is associated with the shape of the boat, which is larger for Athlete A2 ($w = 61$ cm instead of $w = 41$ cm) and, thus, induces a much larger wave drag.

We conclude that the theoretical drag is able to estimate the drag measured experimentally at $\pm 8$%, which is fair considering the different approximations that we have used, especially concerning the shape of the hull.

Concerning the evolution of these different drag contributions with the velocity, we present in Table A1 their values in the range $V \in [1$ m/s–5 m/s]. Equations (A1), (A4) and (A5) are, respectively, used to estimate $F_s$, $F_w$, and $F_a$ using the parameters associated with Athlete A1: $L = 5.2$ m, $w = 0.41$ m, $H = 1.70$ m, and $M_t = 76$ kg.

Even if the different force contributions increase with the velocity, we observe in the last column of Table A1 that the total drag area remains almost constant throughout the whole velocity range $SC_D = 5.7 \pm 0.3 \times 10^{-3}$ m$^2$.

**Table A1.** Evolution of the different drag contributions with the boat velocity estimated using the parameters associated with Athlete A1: $L = 5.2$ m, $w = 0.41$ m, $H = 1.70$ m and $M_t = 76$ kg.

| $V$ (m/s) | $Re_L$ | $Fr$ | $SCs$ (m$^2$) | $F_s$ (N) | $C_w$ | $F_w$ (N) | $F_a$ (N) | $F_D$ (N) | $SC_D$ (m$^2$) |
|---|---|---|---|---|---|---|---|---|---|
| 1 | $0.57 \times 10^7$ | 0.14 | $5.4 \times 10^{-3}$ | 2.7 | $12 \times 10^{-4}$ | 0.1 | 0.16 | 2.95 | $5.9 \times 10^{-3}$ |
| 2 | $1.04 \times 10^7$ | 0.28 | $4.7 \times 10^{-3}$ | 9.5 | $26 \times 10^{-4}$ | 0.9 | 0.62 | 11.0 | $5.5 \times 10^{-3}$ |
| 3 | $1.56 \times 10^7$ | 0.42 | $4.4 \times 10^{-3}$ | 20.0 | $77 \times 10^{-4}$ | 5.8 | 1.4 | 27.2 | $6.0 \times 10^{-3}$ |
| 4 | $2.08 \times 10^7$ | 0.56 | $4.2 \times 10^{-3}$ | 33.8 | $76 \times 10^{-4}$ | 10.2 | 2.5 | 46.6 | $5.8 \times 10^{-3}$ |
| 5 | $2.60 \times 10^7$ | 0.70 | $4.1 \times 10^{-3}$ | 51.0 | $61 \times 10^{-4}$ | 12.8 | 3.9 | 67.2 | $5.4 \times 10^{-3}$ |

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
