# Peer review of "On the Physics of Kayaking"

_applsci, doi:10.3390/app12188925_

Round 1

Reviewer 1 Report

Although the authors made a useful effort in trying characterize the physics of kayak, I have many concerns regarding several aspects of the manuscript. The section where the manuscript was submitted (Mechanical Engineering) receives high-quality papers reporting state-of-the-art technology in the fields of solid mechanics, machine design, advanced manufacturing processes as well as other basic phenomena. In this way I don’t this that this work fits well within that scope.

For me, the work also has serious flaws in the way how it is presented. There is lack of scientific soundness, poor quality of writing and little coherence within and between sections. The introduction is short not covering the state of the art. The research gap also is not clearly defined. There is no hypothesis after the aim of your study. There are also missing aspects about experimental details that need to be provided so that the results can be reproduced. At the end, I ask the author to rethink the way how they want to present their data and, probably, think in other approaches for that.

Author Response

Dear reviewer,

Thank you for your comments and suggestions for the manuscript.

Please find enclosed our reply.

Best

Charlie Pretot for the authors

Reviewer 2 Report

Please find enclosed my review of the manuscript. 

Author Response

(The authors gave the same response as above.)

Reviewer 3 Report

A very interesting topic and one that I am sure is worthy of publication. However, the information is being presented in a manner inconsistent with other scientific research. My best recommendation is to reformat the paper to better reflect a format more common in scientific research. This will enable the consumer to absorb the information more smoothly. There are several areas of the document that need to be checked for grammar and syntax. Spelling errors, floating sentences and confusing presentation of information are found throughout the document. 

Again, I do believe the content of this work is very interesting. The presentation need to be shifted to enable a more fluid consumption of information. 

Please note these suggestions are meant to assist in the crafting of the document. 

Overall, a wonderful effort. 

Author Response

(The authors gave the same response as above.)

Round 2

Reviewer 1 Report

Although I acknowledge the effort shown by the authors in improving the manuscript, I still maintain my previous concerns. So, it falls within the section editor the decision if wants to provide to the readers a publication with many flaws and a lower scientific level This may impair Applied Sci conotation and at last MDPI group. 

Reviewer 3 Report

Well done. 

The additions and clarifications made to the document have improved the readability. 

My suggestions are few, but are believed to enhance the digestibility of the manuscript. 

1. Please be sure the abstract is consistent with the journal's prescribed format. 

2. Perhaps lead the manuscript with your description of kayak and kayak paddling as opposed to a series of figures. 

3. Avoid vague language such as; It or they or people. Be specific when referencing some source or figure. Example: It corresponds, should be - This figure corresponds. Or people have, should be - researchers have. 

4. Make sure the formatting is consistent. Indent all new paragraphs. 

I beleive with minor adjustments this can and will be a very useful paper. 
